# Cytotoxic properties of unfractionated and fractionated bromelain alone or in combination with chemotherapeutic agents in colorectal cancer cells

Kuei-Yen Tsai[1,2,3], Po-Li Wei[1,2,4,5], Mohamed Azarkan[6], Nasiha M'Rabet[6], Precious Takondwa Makondi[7], Hsin-An Chen[2,3], Chien-Yu Huang[8,9]*, Yu-Jia Chang[1,10,11,12]*

1 Graduate Institute of Clinical Medicine, College of Medicine, Taipei Medical University, Taipei, Taiwan, 2 Department of Surgery, School of Medicine, College of Medicine, Taipei Medical University, Taipei, Taiwan, 3 Division of General Surgery, Department of Surgery, Shuang Ho Hospital, Taipei Medical University, New Taipei City, Taiwan, 4 Division of Colorectal Surgery, Department of Surgery, Taipei Medical University Hospital, Taipei Medical University, Taipei, Taiwan, 5 Graduate Institute of Cancer Biology and Drug Discovery, Taipei Medical University, Taipei, Taiwan, 6 Service de Chimie Générale I (CP 609), Faculté de Médicine, Campus Erasme (CP 609), Université Libre de Bruxelles, Brussels, Belgium, 7 Kamuzu Central Hospital—National Cancer Center, Lilongwe, Malawi, 8 School of Medicine, National Tsing Hua University, Hsinchu, Taiwan, 9 Institute of Molecular and Cellular Biology, National Tsing Hua University, Hsinchu, Taiwan, 10 Department of Pathology, Wan Fang Hospital, Taipei Medical University, Taipei, Taiwan, 11 Cell Physiology and Molecular Image Research Center, Wan Fang Hospital, Taipei Medical University, Taipei, Taiwan, 12 Cancer Research Center, Taipei Medical University Hospital, Taipei Medical University, Taipei, Taiwan

* r5424012@tmu.edu.tw (YJC); cyhuang@life.nthu.edu.tw (CYH)

**Data Availability Statement:** All relevant data are within the paper.

## Abstract

### Background

Colorectal cancer (CRC) is one of the most lethal cancers worldwide. Long-term survival is not achieved in metastatic CRC despite the current multidisciplinary therapies. Bromelain, a compound extracted from the pineapple plant, has multiple functions and anticancer properties. Previously, bromelain has been chromatographically separated into four fractions. Fraction 3 (F3) exhibits the highest proteolytic activity. The anticancer effects of F3 bromelain in CRC cells is unknown.

### Methods

In vitro cytotoxicity was verified through a sulforhodamine B assay. Apoptosis in CRC cells induced by unfractionated or F3 bromelain was examined using Annexin V-FITC/PI staining and Western blot analysis. ROS status, autophagy and lysosome formation were determined by specific detection kit.

### Results

The cytotoxicity of F3 bromelain in CRC cells was found to be comparable to that of unfractionated bromelain. F3 bromelain induces caspase-dependent apoptosis in CRC cells. Treatment with unfractionated or F3 bromelain increased superoxide and oxidative stress

**Funding:** The author(s) received no specific funding for this work.

**Competing interests:** The authors have declared that no competing interests exist.

**Abbreviations:** 5-FU, 5-fluorouracil; ATG, autophagy-related gene; BAD, BCL-2-related death agonist; BAX, BCL-2-related X protein; CRC, colorectal cancer; CQ, chloroquine; ERK, extracellular signal-regulated kinase; F3, fraction 3; LC3, light chain 3; MAPK, mitogen-activated protein kinase; mTOR, mammalian target of rapamycin; NF-κB, nuclear factor kappa-light-chain-enhancer of activated B cells; PARP, poly (ADP-ribose) polymerase; PI3K, phosphoinositide 3-kinases; ROS, reactive oxygen species; SRB, sulforhodamine B.

levels and autophagy and lysosome formation. ATG5/12 and beclin-1 were upregulated, and the conversion of LC3B-I to LC3B-II was increased significantly by treatment with F3 bromelain. Treated CQ, autophagy inhibitor, with unfractionated or F3 bromelain enhances the cytotoxic effects. Finally, the combination of unfractionated and F3 bromelain with a routine chemotherapeutic agent (5-fluourouracil, irinotecan, or oxaliplatin) resulted in synergistically higher cytotoxic potency in CRC cells.

## Conclusion

Unfractionated and F3 bromelain inhibits CRC cell proliferation in vitro, and the cytotoxic effects of unfractionated bromelain are equivalent to F3 bromelain. F3 bromelain may be a potential and potent drug for clinical use due to its anticancer efficacy and high synergistic cytotoxicity when combined with a routine chemotherapeutic agent for CRC.

## Introduction

Colorectal cancer (CRC) is the third most common cancer and the second leading cause of cancer-related deaths worldwide [1]. Recent advances in multidisciplinary therapy have led to significant improvements in patient survival rates, but long-term survival is still not being achieved in most patients with metastatic CRC. Current treatments for metastatic CRC include chemotherapy and targeted therapy [2]. Despite advances in treatments, the 5-year overall survival rate for metastatic disease is only 10.5% [3]. Acquired resistance to pharmacological therapy is generally considered to be the main reason for treatment failure. Chemoresistance, which is the most common cause of poor treatment effects in CRC, leads to chemotherapy failure, tumor recurrence, and metastasis. Therefore, the development of new effective or adjunctive therapies for metastatic CRC will have a major effect on patient outcomes.

Modern pharmacology continues to search for novel biologically active compounds with specific functions, especially those without side effects. Recently, interest in plants and natural compounds from plants has increased markedly owing to their relatively high availability, high efficiency, low toxicity, and ease of acquisition. Bromelain is an extract of pineapple plants and a sulfhydryl proteolytic enzyme used in many branches of medicine. Used to treat cardiovascular disease and cancer and in antimicrobial therapy, bromelain also has immunomodulatory effects [4]. The human intestine can absorb bromelain without inducing degradation or loss of its biological properties [5]. Because of its antitumor effects, possible mechanisms have been identified in previous research, including modulation of the gene expression of cell proliferation, induction of cell apoptosis and autophagy, and inhibition of cyclins to arrest the cell cycle [4]. Bromelain is an extract compound containing many enzymes, including thiol proteases, phosphatases, cellulases, peroxidases, and glucosidases. Previously, bromelain was fractionated in the hope that specific enzymatic functions or therapeutic properties would be confined exclusively to its fractions [6]. We believe that these fractions may not have the same proteolytic potency and anticancer effects as bromelain itself. To explore the relevant mechanisms in detail and develop applicable therapeutic agents, the exclusive functions of purified fractionated bromelain must be investigated.

Badar et al. chromatographically separated bromelain into four fractions and examined their proteolytic, anticancer, and antithrombotic activities. Fraction 3 bromelain (F3 bromelain), the highly active basic bromelain isoform, had the highest proteolytic and anticancer activities [7]. In combination with chemotherapeutic agents, F3 bromelain exhibited greater

treatment efficacy than did unfractionated bromelain in pancreatic and hepatocellular carcinoma cell lines [7]. Our previous study revealed that bromelain resulted in considerable oxidative stress and the production of superoxides in CRC cells followed by the triggering of autophagy, causing cell death and inhibiting carcinogenesis [8]. However, whether the cytotoxic effects of unfractionated and F3 bromelain have the same regulatory mechanisms in CRC is unclear. In this study, the anticancer properties of F3 bromelain were compared with those of unfractionated bromelain. The anticancer properties of unfractionated bromelain were discovered to be equivalent to F3 bromelain. F3 bromelain could serve as a potential adjunctive therapy for CRC.

## Materials and methods

### Fractionation of ananas comosus stem proteases

Stem bromelain proteases (Sigma-Aldrich, Ref. B4882; 3 units/mg protein) were fractionated as described previously [6]. Briefly, stem bromelain powder was suspended in 100 mM sodium acetate buffer (pH 5.0) in the presence of S-methyl methanethiosulfate (MMTS, Sigma-Aldrich, Ref. 64306), a reversible thiol-blocking reagent, under constant and moderate stirring for 1 h at 4˚C. MMTS was added to prevent protease autolysis and irreversible oxidation of the catalytic cysteine. The obtained suspension was then ultracentrifuged (35,000 × g, 4˚C, 30 min), and the supernatant corresponding to the total soluble protein fraction was applied to a homemade SP-Sepharose Fast Flow column (13 cm × 2.5 cm internal diameter; GE Healthcare, Uppsala, Sweden) pre-equilibrated with three column volumes of 100 mM sodium acetate buffer (pH 5.0) and eluted with a linear concentration gradient of sodium acetate buffer (pH 5.0) from 100 to 800 mM (total volume 1700 mL, 60 mL/h). The flow-through material was washed away with 10 column volumes of the pre-equilibrating buffer. The amidase activity of the fractions was performed with the substrate DL-BAPNA (Sigma-Aldrich, Ref. B4875), as previously reported [6]. The fractions corresponding to basic bromelain were pooled in accordance with their amidase activity profile. In a previous study, we demonstrated that F3 bromelain contained basic bromelain and a lectin [7]. To purify basic bromelain, we used D-mannose-agarose as an affinity support as described in [7]. The flow-through fraction, containing basic bromelain, was further submitted to cation-exchange chromatography on a HiTrap SP-Sepharose Fast Flow column (5 cm, VWR) coupled to an AKTAprime plus fast protein liquid chromatography device using the following conditions: pressure limit 0.5 MPa, three column volumes of 100 mM pre-equilibrating sodium acetate buffer (pH 5.0), three column volumes of washing buffer (100 mM sodium acetate buffer, pH 5.0), elution with a linear concentration gradient of sodium acetate buffer (pH 5.0) from 100 to 800 mM (total volume 100 mL, 1 mL/min). The fractions corresponding to basic bromelain were concentrated through ultrafiltration and activated for 15 min at 37˚C with 10 mM final concentration L-cysteine (BioUltra ≥ 98.5% (RT) Merck, Ref. 30089-25G), after which they were immediately exhaustively dialyzed overnight against water at 4˚C. After dialysis, the sample was lyophilized and stored at −20˚C until use.

### Chemicals, reagents, and cell cultivation

The bromelain and all other reagents used in the study were purchased from Sigma Chemical (St. Louis, MO, USA). The bromelain was fractionated as previously described [6]. Human colon adenocarcinoma cell lines—DLD-1 (CCL-221), HT29 (HTB-38), and HCT116 (CCL-247)—were obtained from the American Type Culture Collection (Rockville, MD, USA). The cells were grown in RPMI 1640 medium supplemented with 10% fetal bovine serum, 100 IU/mL penicillin, and streptomycin (100 μg/mL) and incubated at 37˚C in air containing 5% $CO_2$.

## Sulforhodamine B colorimetric assay

The cells ($2 \times 10^4$) were cultivated in 24-well plates and incubated overnight, after which they were treated with bromelain fractions, unfractionated bromelain, or a control (distilled $H_2O$) for 48 h. After the incubation period, the cells were fixed with 10% (w/v) trichloroacetic acid, stained with protein-bound sulforhodamine B (SRB) for 30 min, and then washed repeatedly with 1% (vol/vol) acetic acid to remove excess dye. The protein-bound dye was dissolved in 10 mM Tris base solution, and the optical density was then determined at 515 nm by using a microplate reader. The cells treated with the control were defined as the baseline, and fold change as the optical density value of cells treated with unfractionated or F3 bromelain was calculated relative to the baseline.

## Total reactive oxygen species and superoxide detection by using the FlexiCyte protocol

An ROS-ID Total ROS/Superoxide Detection Kit (ENZ-51010, Madison, NY, USA) was used in accordance with the manufacturer's protocol to detect reactive oxygen species (ROS) levels. Briefly, cells ($2.4 \times 10^5$) were seeded in six-well plates overnight and cultivated with unfractionated or F3 bromelain, or the vehicle for 24 and 48 h. After treatment, the cells were harvested and stained with two fluorescent dyes, green dye (total ROS detection reagent) and orange dye (superoxide detection reagent), to detect the levels of real-time oxidative stress and superoxides in living cells, respectively. Hoechst-33342 was used to stain harvested cells, enabling detection of the total cell population. The intensity of the fluorescent dye (indicating the number of cells) was measured using the NucleoCounter NC-3000 system (ChemoMetec A/S, Allerod, Denmark).

## Autophagy and lysosome detection by the FlexiCyte protocol

The Cyto-ID Autophagy Detection Kit (ENZ-51031) and Lyso-ID Green Detection Kit (ENZ-51034) were used in accordance with the manufacturer's protocol to detect autophagy and lysosomes, respectively. Cells ($2.4 \times 10^5$) were cultured in six-well plates overnight and incubated with unfractionated or F3 bromelain, or the control for 24 and 48 h. After this treatment, the cells were stained with Cyto-ID fluorescent dyes to detect autophagic vacuoles, including preautophagosomes, autophagosomes, and autolysosomes. Similarly, cells were stained using the Lyso-ID Green Detection Kit, which measures the level of lysosome formation. Hoechst-33342 was employed to stain harvested cells and thus enable detection of the total cell population. The intensity of the fluorescent dye (indicating the number of cells) was measured using the NucleoCounter NC-3000 system (ChemoMetec A/S, Allerod, Denmark).

## Reverse transcription–polymerase chain reaction and quantitative reverse transcription–polymerase chain reaction analysis

The cultured cells were extracted using TRIZOL reagent (Invitrogen Life Technologies, Carlsbad, CA, USA). Total RNA (8 μg) was used for the reverse transcription reaction in a 20-μL reaction volume to synthesize cDNA by using a cDNA Synthesis Kit (Invitrogen Life Technologies). Reverse transcription was performed in accordance with the manufacturer's instructions. Real-time polymerase chain reaction was performed using the ABI SYBR Green Master Mix (Applied Biosystems). Thermal cycling was performed using an ABI 7500 FAST instrument. Each sample was run in triplicate for each experiment, and each experiment was performed thrice. The expression levels of the target genes were normalized to that of glyceraldehyde 3-phosphate dehydrogenase.

## Annexin V-FITC/propidium iodide assay

Cells were seeded in six-well plates overnight and treated with unfractionated or F3 bromelain and vehicle control for 48 h. The cells were then centrifuged and washed with phosphate-buffered saline, and the cell pellets were resuspended in 100 μL of binding buffer. An Annexin V-FITC Apoptosis Detection Kit (Cat No.: AVK250, Strong Biotech Corporation) was used in accordance with the manufacturer's instructions to calculate the number of apoptotic cells. The cells were stained with Annexin V and propidium iodide for 15 min in the dark. Apoptosis was examined through flow cytometry (BD Biosciences, San Diego, CA, USA).

## Protein extraction and Western blot analysis

Cells were treated with unfractionated or F3 bromelain and the vehicle control for 48 h. Cell lysates were subjected to sodium dodecyl sulfate–polyacrylamide gel electrophoresis and electrotransferred onto polyvinylidene difluoride transfer membranes (GE Healthcare, Piscataway, NJ, USA) for antibody blotting. The membranes were incubated with antibodies against autophagy-related gene (ATG)5, ATG12, beclin-1, light chain 3 (LC3), cleaved (c)-caspase-3, c-caspase-8, c-caspase-9, or poly(ADP-ribose) polymerase (PARP) at 4˚C overnight, after which they were probed with secondary antibodies individually for 1 h. Enhanced chemiluminescence reagent (GE Healthcare Piscataway, NJ, USA) and VersaDoc 5000 (Bio-Rad Laboratories, Hercules, CA, USA) were used to detect the target bands.

## Statistical analysis

Data were analyzed using Microsoft Excel. Values are presented as the mean ± standard deviation (SD) of at least three independent experiments. Differences in continuous variables between groups were examined using Student's t test or the Mann–Whitney U test. For 50% inhibitory concentration ($IC_{50}$) experiments, two groups were compared using one-way analysis of variance or the two-tailed unpaired t test. Statistical significance was indicated by $p < 0.05$ (*, $p < 0.05$; **, $p < 0.01$).

# Results

## Cytotoxicity of unfractionated bromelain for CRC cells is confined exclusively to F3 bromelain

Our previous study revealed that the survival of CRC cells is reduced when they were treated with unfractionated bromelain [8]. In the present study, we first investigated the cytotoxic properties of unfractionated or F3 bromelain. As shown in Fig 1, treatment with unfractionated or F3 bromelain significantly reduced the survival and proliferation of CRC cells—the DLD-1, HT29, and HCT116 cell lines. The $IC_{50}$ values of unfractionated and F3 bromelain were approximately 20 μg/mL for HCT 116, 40 μg/mL for DLD-1 and HT29. These results indicate that the ability of F3 bromelain to inhibit CRC cells is comparable to that of unfractionated bromelain, suggesting that the cytotoxic effects of bromelain are confined exclusively to F3 bromelain.

## F3 bromelain induces apoptosis in CRC cells

Apoptosis, programmed cell death, plays a critical role in tumor suppression [9]. Our previous study revealed that treatment with unfractionated bromelain induces apoptosis in CRC cells. The effect of F3 bromelain on caspase-dependent apoptosis was examined. Flow cytometry demonstrated that the ratio of apoptosis was significantly higher in the CRC cells treated with

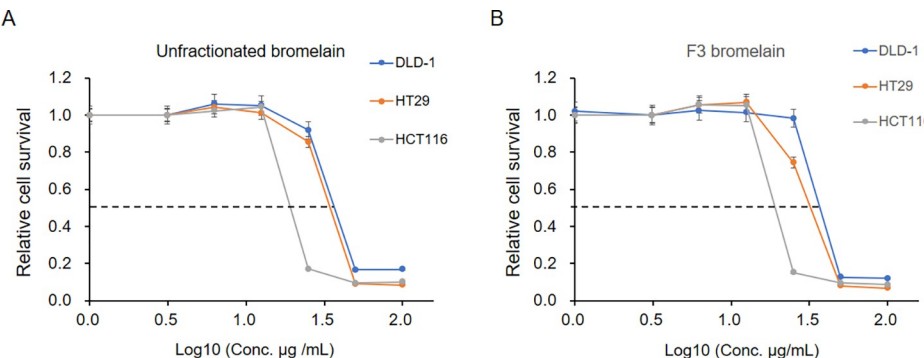

**Fig 1. Unfractionated and F3 bromelain exhibited cytotoxicity in colorectal cancer cells.** After treatment of DLD-1, HT29, and HCT116 cell lines with unfractionated or F3 bromelain at different doses (0 to 100 μg/mL), the relative cell survival rate was determined through an SRB colorimetric assay. Unfractionated bromelain treatment (**A**) and F3 bromelain (**B**) resulted in reduced cell survival in colorectal cancer cells in a dose-dependent manner compared with the survival of the control-treated cells, which was defined as 100%. X-axis represents the concentration with $Log_{10}$ (Concentration).

unfractionated or F3 bromelain than in those treated with the vehicle (Fig 2A–2C). Similarly, Western blot analysis revealed that upregulation of c-PARP, c-caspase 3, c-caspase 8, c-caspase 9, and proapoptotic members BCL-2-related X protein (Bax) and BCL-2-related death agonist (Bad) was greater in the CRC cells treated with unfractionated or F3 bromelain than in those treated with vehicle (Fig 2D and 2E). Collectively, these results indicate that unfractionated or F3 bromelain treatment induces caspase-dependent apoptosis in CRC cells.

## F3 bromelain induces oxidative stress and superoxide generation

ROS play an important role in mediating cancer cell death. ROS-dependent cell death pathways are involved in the mechanisms of a number of therapeutic agents [10]. 5-Fluorouracil (5-FU), a conventional chemotherapeutic agent used for the treatment of CRC, induces Src activation through ROS pathways [11]. A previous study showed that treatment with bromelain increased ROS levels in breast carcinoma cells [12], and another study showed that treatment with bromelain increased ROS levels in CRC cells. In the present study, ROS levels were detected using an ROS/Superoxide Detection Kit. ROS levels and superoxide production were markedly higher in the CRC cells treated with unfractionated or F3 bromelain than in those treated with the vehicle control (Fig 3A). Furthermore, mRNA expression of the antioxidant genes peroxiredoxin1 (*PRDX1*) and *PRDx4* was significantly lower in the CRC cells treated with unfractionated or F3 bromelain than in those treated with the vehicle control (Fig 3B). Taken together, these data indicate that treatment with unfractionated or F3 bromelain causes cytotoxic effects in CRC cells by inducing ROS and superoxide production.

## F3 bromelain increases autophagy and lysosomal formation in CRC cells

Autophagy is a process that involves the recycling of nutrients and the generation of energy through selective degradation of cellular components and is considered a cell survival mechanism [13]. Autophagy is an essential mechanism that plays a key role in cancer suppression and progression. Researchers have demonstrated that autophagy is induced by ROS [14, 15]. To determine whether F3 bromelain upregulates the activation of autophagic processes, autophagic vacuoles and autophagic flux were examined using a Cyto-ID Autophagy Detection Kit. Levels of autophagy were higher in the CRC cells treated with unfractionated or F3 bromelain than in those treated with the vehicle control (Fig 4A). Similarly, lysosome formation was

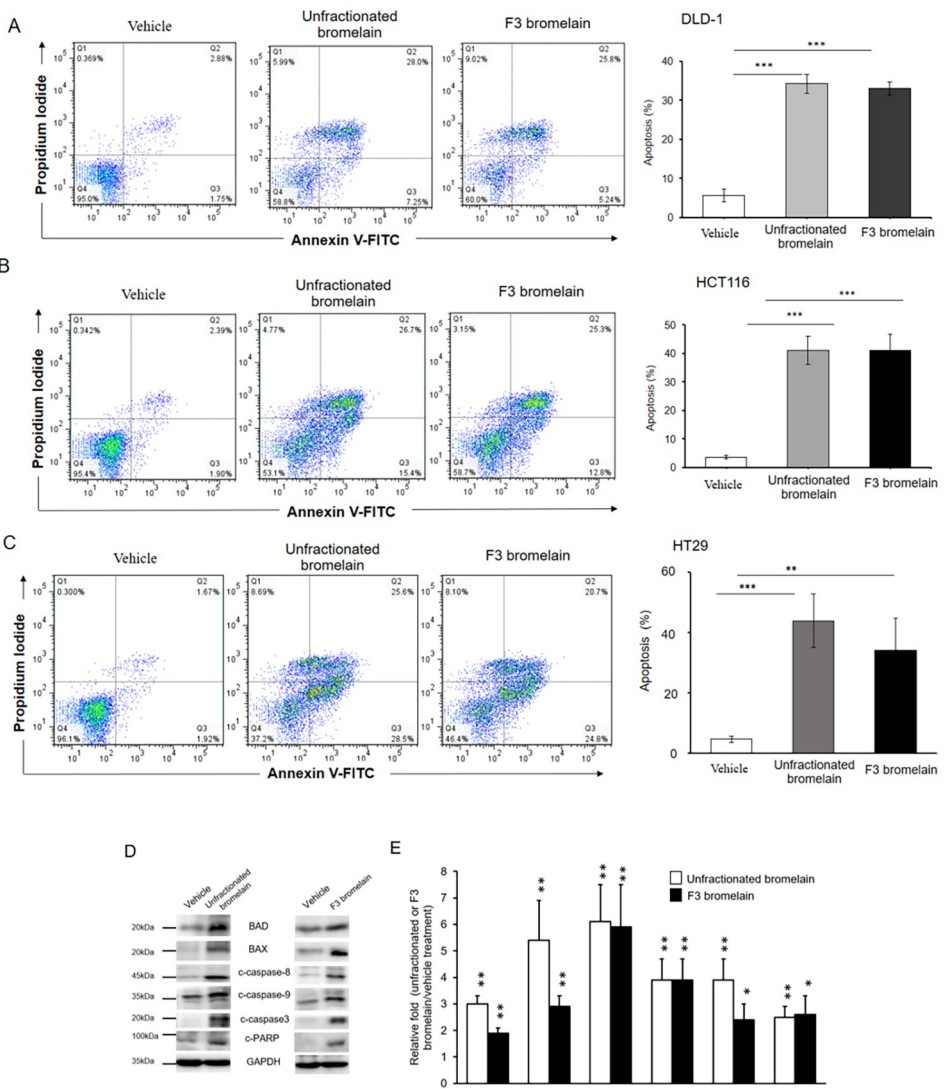

**Fig 2. Unfractionated and F3 bromelain treatment induced apoptosis in colorectal cancer cells. (A–C)** The apoptotic cell population was analyzed using Annexin V-FITC/propidium iodide staining and FACS analysis in DLD-1, HCT116, and HT29 cells after being treated with unfractionated or F3 bromelain. The apoptosis cell population was indicated by the percentage of Q2+Q3. **(D)** Protein levels of apoptosis-related genes were determined through Western blot analysis. Cells treated with unfractionated or F3 bromelain exhibited increased levels of Bax, Bad, cleaved (c)-caspase 3, c-caspase 8, c-caspase 9, and c-PARP. All experiments were performed at least three times independently (* $p < 0.05$, ** $p < 0.01$, *** $p < 0.001$).

greater in the CRC cells treated with unfractionated or F3 bromelain than in those treated with the vehicle control (Fig 4B), where lysosome formation was measured using a Lyso-ID Green Detection Kit. Western blot analysis was performed to further verify the expression of autophagy-related proteins, including ATG5/12, LC3B-I, LC3B-II, and beclin-1. ATG5/12 and beclin-1 were discovered to be upregulated, and the conversion of LC3B-I to LC3B-II was increased significantly by treatment with unfractionated or F3 bromelain (Fig 4C). These data suggest that treatment with F3 bromelain induces autophagy and lysosome formation in CRC cells.

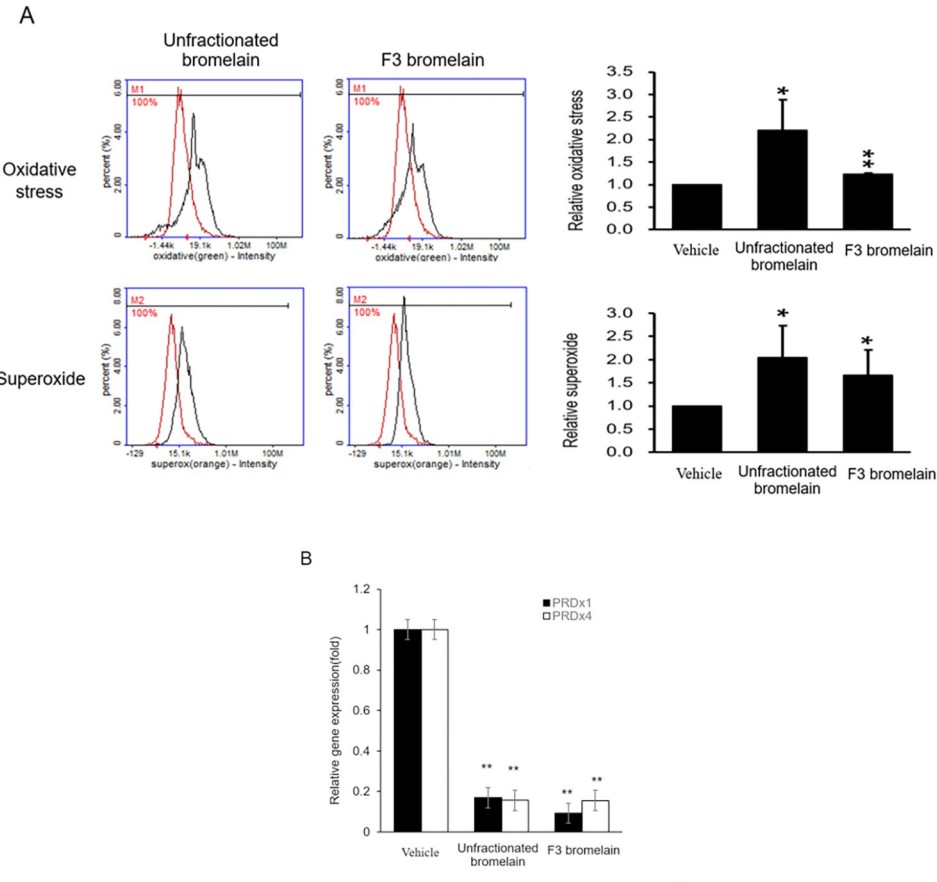

**Fig 3. Unfractionated and F3 bromelain increased reactive oxygen species (ROS) levels and superoxide production in colorectal cancer cells.** HCT116 cells were treated with unfractionated bromelain at 15 μg/mL or F3 bromelain at 15 μg/mL for 48 h. Compared with the control, unfractionated or F3 bromelain treatments significantly increased ROS and superoxide levels, which were detected using a fluorescent dye detection kit (**A**). Relative mRNA levels of antioxidant genes were lower after treatment with unfractionated or F3 bromelain (**B**) (* $p < 0.05$, ** $p < 0.01$).

## Autophagy plays a cytoprotective role in CRC cells treated with unfractionated or F3 bromelain

The dual role of autophagy in tumor promotion and suppression in different cancer types and stages has been well established [16]. One study found that CRC cells treated with chloroquine (CQ) are sensitized to antiangiogenesis and chemotherapy drugs [17]. CQ inhibits autophago-some–lysosome fusion. We treated CQ to block the autophagy with unfractionated or F3 bromelain treatment to determine the role of autophagy in bromelain-induced cell death. As shown in Fig 5A, cell viability was not altered by CQ treatment alone, however, in the combination treatment in unfractionated or F3 bromelain with CQ showed a decrease of cell viability compared to unfractionated or F3 bromelain treatment only. In addition, Western blot analysis indicated that LC3B-II protein levels were significantly increased by the unfractionated or F3 bromelain treatment with CQ, indicating that autophagy flux was blocked, which caused the accumulation of LC3B-II (Fig 5B). Collectively these results demonstrate that F3 bromelain treatment induces autophagy in CRC cells and that inhibition of autophagy enhances the cytotoxic effects of F3 bromelain. Those results indicate that bromelain-induced autophagy plays a protective role in CRC cells.

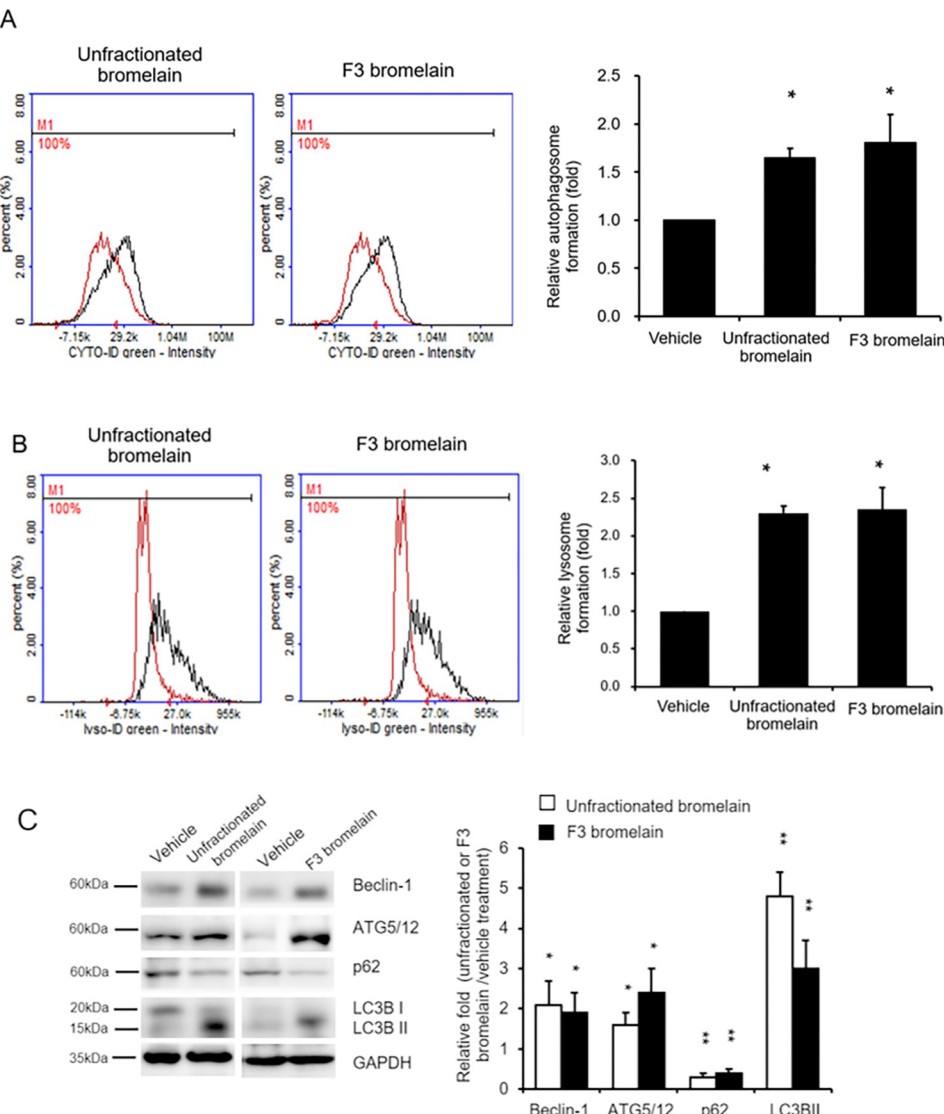

**Fig 4. Unfractionated and F3 bromelain induced autophagy and lysosome formation in colorectal cancer cells.**
HCT116 cells were treated with unfractionated bromelain at 15 µg/mL or F3 bromelain at 15 µg/mL for 48 h, and autophagic vacuoles and autophagic flux were detected using a Cyto-ID Autophagy Detection Kit (**A**). Lysosome formation was detected using a Lyso-ID Green Detection Kit (**B**). Autophagosome and lysosome formation was increased after the treatment. Relative levels of autophagy-related proteins were obtained through Western blot analysis, and the results indicated that autophagy was induced after treatment with unfractionated or F3 bromelain (**C**) (* $p < 0.05$).

## Synergistic effects of F3 bromelain and routine chemotherapeutic agents in CRC

One study revealed that F3 bromelain exhibits greater treatment efficacy in pancreatic and liver cancer cells when it is combined with chemotherapeutic agents [7]. We verified the cytotoxic effect of unfractionated or F3 bromelain combined with chemotherapeutic agents (5-FU, irinotecan, and oxaliplatin) on CRC tumor suppression to determine whether any synergistic effects existed. The reduction in cell viability was significantly greater in the cells treated with unfractionated or F3 bromelain combined with 5-FU than in the cells treated with 5-FU alone (Fig 6). Combination therapy of unfractionated or F3 bromelain with irinotecan or oxaliplatin

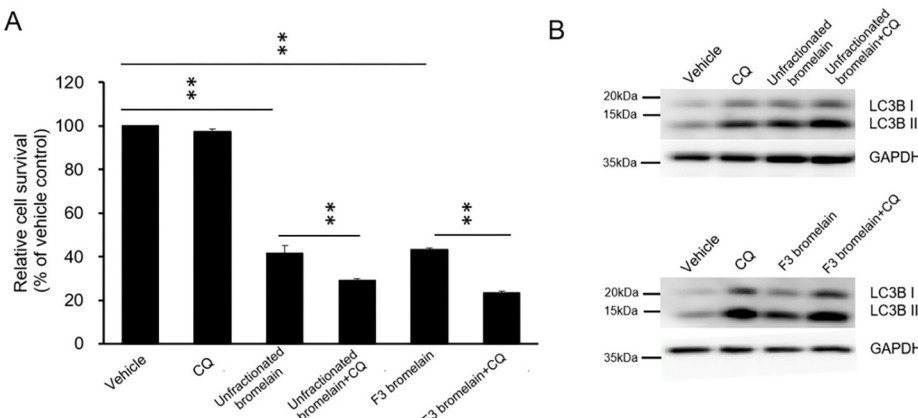

**Fig 5. Blocking autophagy further enhanced unfractionated bromelain–induced or F3 bromelain–induced cytotoxicity in colorectal cancer cells.** The viability of CRC cells treated with unfractionated or F3 bromelain, in the presence or absence of chloroquine (CQ), was determined through an SRB colorimetric assay (**A**). Cell viability was further decreased by autophagy blockage. Autophagy blockage was confirmed by accumulation of LC3-II in Western blot analysis (**B**) (* p < 0.05, ** p < 0.01).

also had clear synergistic effects (Fig 6). Cytotoxicity in CRC cells was greater after the addition of unfractionated or F3 bromelain. The combination of unfractionated or F3 bromelain with a routine chemotherapeutic agent has a synergistic effect, resulting in superior cytotoxicity in CRC cells.

## Discussion

CRC is a highly prevalent cancer with unsatisfactory prognosis and becoming more common owing to modern dietary habits and lifestyles. Therefore, powerful drugs that work effectively and efficiently through various antitumor mechanisms are urgently needed. Various plant-based natural compounds have been reported to have therapeutic effects, and some of these metabolites can act as antioxidants and antitumor drugs owing to their cytotoxic effects [18, 19]. Bromelain, found in pineapple juice and the stem of the pineapple plant, is a digestive

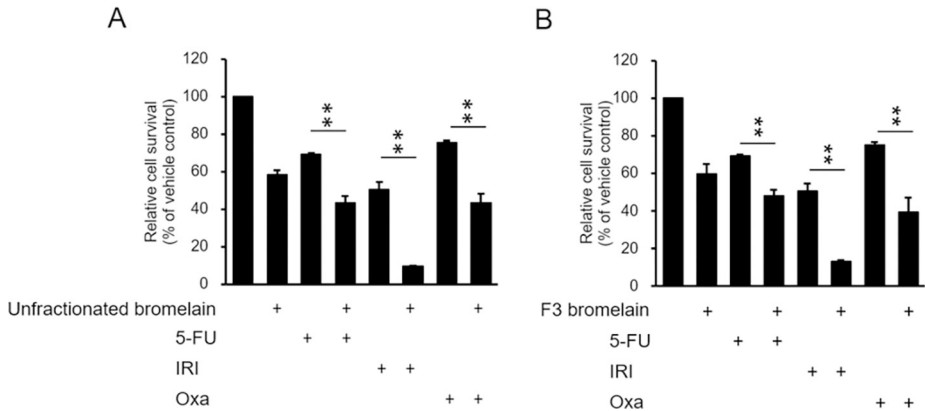

**Fig 6. Combination treatment with unfractionated or F3 bromelain further enhances chemotherapy-induced cytotoxicity in colorectal cancer.** Cytotoxic effect of unfractionated bromelain at 15 μg/mL (A) as well as F3 bromelain at 15 μg/mL (B) in combination with routine chemotherapeutic agents (5-FU at 12.5 μM, irinotecan at 40μM, or oxaliplatin at 2.5 μM) was evaluated. The results showed synergistic cytotoxicity of unfractionated or F3 bromelain in combination with 5-FU, irinotecan, or oxaliplatin (* p<0.05, ** p<0.01).

enzyme that remains active in both the acidic environment of the human stomach and the alkaline environment of the small intestine. Studies have shown that bromelain appears to be absorbed from the intestine at an absorption rate of approximately 40% and that it has a half-life of 6 to 9 hours and remains biologically active [5, 20]. Bromelain is a complex of active compounds with multiple functions. To apply bromelain correctly and accurately, further exploration of its mechanism of action and the extraction of single compounds are necessary. This study examined the anticancer effects of F3 bromelain on CRC cells, and the results indicate that the cytotoxicity of F3 bromelain is comparable to that of unfractionated bromelain; thus, the anticancer effects may be exclusively confined to F3 bromelain. It may be possible to further identify and analyze the single molecular species contained in bromelain that have anticancer effects and to investigate the optimal dosage or route of administration and the possible interactions with other therapeutic drugs for cancer treatment.

ROS play a double-edged role in cancer proliferation. The regulation of ROS is critical for cell viability and proliferation [21]. The carcinogenic roles of ROS include genetic alterations, cell proliferation through the PI3K/AKT/mTOR and MAPK/ERK pathways, and promotion of the epithelial–mesenchymal transition, whereas their tumor-suppressive roles include apoptosis, necroptosis, and ferroptosis [22]. Bromelain has been reported to have considerable anticancer potential by causing intracellular glutathione depletion, ROS production, and the subsequent depolarization of mitochondrial membranes. Bromelain induces cell cycle arrest at the G2/M phase with the concomitant induction of apoptosis [23]. Bromelain also has radio-sensitizing effects and can significantly reduce tumor volume and PARP-1, Ki-67, and NF-κB levels while significantly increasing ROS content and lipid peroxidation in breast carcinoma cells [12]. Our results consistently indicated that oxidation levels and superoxide production were significantly increased by treatment with F3 bromelain and that the anticancer effects were mediated by ROS-dependent cell death. Interestingly, bromelain treatment was reported to inhibit CRC cell proliferation and decrease ROS production [24]. This implies that bromelain has anticancer effects, but its regulatory effect on ROS remains to be clarified.

Studies have demonstrated that autophagy and apoptosis are induced by intracellular ROS [25, 26]. Autophagy plays a crucial role in cancer cells in harsh environments, such as those resulting from oxidative stress. However, to help clear away damaged cells and preserve cellular integrity, unresolved stress beyond tolerance limits can lead to cell death [27]. Autophagy is generally considered to have a tumor-suppressive role because autophagy-related genes are frequently defective in human tumors. Autophagic dysfunction results in increased levels of oxidative stress and is associated with genomic instability, tumorigenesis, and malignant transformation [28]. Our previous report also showed increased autophagosome and lysosome formation after treatment with bromelain [8]. However, once cancer has developed, autophagy has the opposite role and contributes to tumor cell survival and progression [29]. Autophagy can also enhance tumor cell survival under conditions of metabolic stress and subsequently promote chemoresistance [30]. In the present study, the results consistently demonstrated that autophagy was induced in CRC cells treated with F3 bromelain and that autophagy promoted tumor progression by inhibiting the cytotoxicity of F3 bromelain.

Bromelain has been used clinically for more than 50 years in many branches of medicine. Previous studies have attempted to fractionate bromelain and have characterized its fractionated forms [31–33]. However, few studies have investigated the potential clinical applications of fractionated bromelain. Bromelain remains a topic of interest because it has multiple functions, low toxicity, and high therapeutic efficacy. Recently, Matagne et al. purified and fractionated bromelain for detailed physicochemical and enzymatic characterization [6], and the proteolytic, cytotoxic, and anticoagulant properties of various fractionated forms were verified and compared [7]. Nevertheless, studies on the anticancer effects of fractionated bromelain

remain scarce. The present study may be the only study to demonstrate that F3 bromelain has a potent cytotoxic effect comparable to that of unfractionated bromelain in CRC. This implies that the potent cytotoxic effects of bromelain are confined exclusively to F3. Exploring the numerous roles of bromelain at the molecular level remains difficult, and more detailed studies involving antitumor and proteomic analyses are warranted to identify potential specific targets affected by bromelain. Our study suggests the possibility of precise separation of single molecules and further dissection of the anticancer mechanism in bromelain.

In conclusion, our results show that, in vitro, F3 bromelain exhibits anticancer effects equivalent to those of unfractionated bromelain. F3 bromelain treatment activated apoptosis by increasing the production of ROS, which subsequently inhibited the proliferation of CRC cells and induced autophagy. We also revealed that the activation of autophagy is a protective response against bromelain-induced cell death in CRC cells. These findings suggest that F3 bromelain may be a promising adjunctive drug for the treatment of CRC and is worthy of further investigation to identify the specific molecules of interest in bromelain and may help understanding their mechanism of action.

## Author Contributions

**Conceptualization:** Kuei-Yen Tsai, Po-Li Wei, Mohamed Azarkan, Nasiha M'Rabet, Chien-Yu Huang, Yu-Jia Chang.

**Data curation:** Kuei-Yen Tsai, Po-Li Wei, Mohamed Azarkan, Precious Takondwa Makondi, Hsin-An Chen.

**Formal analysis:** Hsin-An Chen.

**Funding acquisition:** Yu-Jia Chang.

**Investigation:** Kuei-Yen Tsai, Mohamed Azarkan, Nasiha M'Rabet, Precious Takondwa Makondi, Yu-Jia Chang.

**Methodology:** Nasiha M'Rabet, Precious Takondwa Makondi, Hsin-An Chen, Chien-Yu Huang.

**Project administration:** Po-Li Wei, Chien-Yu Huang, Yu-Jia Chang.

**Resources:** Po-Li Wei, Mohamed Azarkan, Nasiha M'Rabet, Chien-Yu Huang, Yu-Jia Chang.

**Software:** Po-Li Wei, Mohamed Azarkan.

**Supervision:** Yu-Jia Chang.

**Validation:** Kuei-Yen Tsai, Mohamed Azarkan, Hsin-An Chen, Chien-Yu Huang.

**Writing – original draft:** Kuei-Yen Tsai, Mohamed Azarkan, Precious Takondwa Makondi, Hsin-An Chen, Chien-Yu Huang, Yu-Jia Chang.

**Writing – review & editing:** Yu-Jia Chang.

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
