## [Decision Letter · Decision Letter 0]

29 Mar 2023

PONE-D-23-03463Cytotoxic Properties of Unfractionated and Fractionated Bromelain Alone or in Combination with Chemotherapeutic Agents in Colorectal Cancer CellsPLOS ONE

Dear Dr. Chang,

Thank you for submitting your manuscript to PLOS ONE. After careful consideration, we feel that it has merit but does not fully meet PLOS ONE’s publication criteria as it currently stands. Therefore, we invite you to submit a revised version of the manuscript that addresses the points raised during the review process.

We look forward to receiving your revised manuscript.

Kind regards,

Sreeparna Banerjee, PhD

Academic Editor

PLOS ONE

Additional Editor Comments:

Both reviewers have identified several points for further clarification, including how bromelain itself contributes towards cytotoxicity, details on the methodology and the role of the lectin moeity.

Reviewers' comments:

Reviewer's Responses to Questions

**Comments to the Author**

1. Is the manuscript technically sound, and do the data support the conclusions?

Reviewer #1: Yes

Reviewer #2: Partly

2. Has the statistical analysis been performed appropriately and rigorously? 

Reviewer #1: Yes

Reviewer #2: Yes

3. Have the authors made all data underlying the findings in their manuscript fully available?

Reviewer #1: Yes

Reviewer #2: Yes

4. Is the manuscript presented in an intelligible fashion and written in standard English?

Reviewer #1: Yes

Reviewer #2: Yes

5. Review Comments to the Author

Reviewer #1: 1. When F3 bromelain has shown the cytotoxic properties with that of unfractinated bromelain, what is the point in fractionating bromelain?

2. Since Bromelain is a complex of proteolytic enzymes, it is necessary to identify single molecule or individual proteolytic enzyme involved in cytotoxicity in CRC cells. How do the authors justify this?

3. The study has evaluated F3 bromelain in combination with routine chemotherapeutic agents and found out that synergistic cytotoxicity. In this evaluation, how quantitatively F3 bromelain contributes to cytotoxicity?

4. Methods: Is it correct that authors have not isolated bromelain naturally from pineapple, instead they have used commercially available bromelain from Sigma-Aldrich?.

Reviewer #2: The MS deals with the purification of an active fraction from bromelain and the comparison of anti tumor effects with unfractionated sample, using colorectal cancer cells.

The topic is interesting and follows some other papers (i.e. Badar et al 2021, PMID: 34150016).

However, in my opinion the MS is very difficult to be read. I suggest to join Results and Discussion sections. The splitting make the results more difficult to be interpreted. Moreover the Discussion is too long, repeating a lot of information from Introduction.

So, in my opinion, during the merging of these two sections the authors should shorten these parts, and conversely better highlight the relationships between their results and the conclusions that they draw.

In the present form it is very difficult to appreciate the MS and focus on the Results presented.

The authors state that F3 contains a lectin: did they check the hemagglutination activity of the fraction?

MINOR POINTS:

Page 4: please, insert more data about chromatographic runs (for example flow rate and CV for each step).

Page 9 line 28: “chromatographically”.

Figure 1: uL (capitol L)

Figure 2 is very difficult to be read.

6. PLOS authors have the option to publish the peer review history of their article (what does this mean?). If published, this will include your full peer review and any attached files.

Reviewer #1: No

Reviewer #2: No

---

## [Author Response · Author response to Decision Letter 0]

20 Apr 2023

Response to the critiques:

Reviewer #1: 

Q1. When F3 bromelain has shown the cytotoxic properties with that of unfractionated bromelain, what is the point in fractionating bromelain?

Response: Bromelain is a long-established drug with high absorption rate, safety, and few side effects. However, it is a complex of a group of enzymes. In such a drug we cannot know which molecule its special curative effect comes from, and it cannot be further purified to maximize its efficacy. Because previous studies have found that bromelain has anti-tumor properties, the purpose of chromatographic fractionation is to try to further separate individual parts, and to find out the part that really has a curative effect. It is better to perform a study with a homogeneous and well-characterized fraction than with a mixture containing proteases, other proteins and non-protein compounds. The interpretation of the results and sample standardization will be easier when using a well-characterized sample. A previous study has found that the combined use of F3 and chemotherapeutic drugs has a more significant inhibitory effect on pancreatic cancer and HCC than the combination of bromelain and chemotherapeutic drugs (Ref 7). The results in the current study indicate that the cytotoxicity of F3 bromelain on CRC is comparable to that of unfractionated bromelain; thus, the anticancer effects may be exclusively confined to F3 bromelain. In addition, we can check the contribution of the proteolytic activity to the cytotoxic effect by using specific inhibitors of basic bromelain (F3 fraction). This will be difficult to assess by using the complex mixture of stem bromelain. 

Q2. Since Bromelain is a complex of proteolytic enzymes, it is necessary to identify single molecule or individual proteolytic enzyme involved in cytotoxicity in CRC cells. How do the authors justify this?

Response: Stem bromelain contains, in addition to non-protein compounds and non-proteolytic enzymes, at least eight cysteine proteases of the papain family (Ref 6, Matagne et al. 2017, Phytochemistry 138: 29-51). We already demonstrated that these proteases display different substrate specificity and inhibitory properties. We assessed the different fractions on cytotoxicity in CRC cells and the results clearly showed that the fraction containing basic bromelain (F3) gives the expected cytotoxic properties. The reason why we opted for the use of basic bromelain (F3) in our study.

Q3. The study has evaluated F3 bromelain in combination with routine chemotherapeutic agents and found out that synergistic cytotoxicity. In this evaluation, how quantitatively F3 bromelain contributes to cytotoxicity?

Response: Bromelain is a compound with multiple proteolytic enzymes and has diverse functions, but its cytotoxic effect on tumor cells is still unclear. Our current research points out that Bromelain/F3 can induce apoptosis in CRC cells, increase oxidative stress, and regulate autophagy. SRB assay in the current study detected that Bromelain/F3 has 50% inhibitory ability to the growth of CRC cells at 20~40 μg/mL.

Q4. Methods: Is it correct that authors have not isolated bromelain naturally from pineapple, instead they have used commercially available bromelain from Sigma-Aldrich?.

Response: Yes, we used a commercially available bromelain from Sigma-Aldrich.

Reviewer #2: 

Q1. The MS deals with the purification of an active fraction from bromelain and the comparison of antitumor effects with unfractionated sample, using colorectal cancer cells.

The topic is interesting and follows some other papers (i.e. Badar et al 2021, PMID: 34150016).

However, in my opinion the MS is very difficult to be read. I suggest to join Results and Discussion sections. The splitting makes the results more difficult to be interpreted. Moreover the Discussion is too long, repeating a lot of information from Introduction.

So, in my opinion, during the merging of these two sections the authors should shorten these parts, and conversely better highlight the relationships between their results and the conclusions that they draw.

In the present form it is very difficult to appreciate the MS and focus on the Results presented.

Response: Fractionation of bromelain has been reported in a previous study (Ref 7). Therefore we cannot present repetitive results in the current study. The focus of this study is on the anti-tumor effects in CRC comparing F3 and unfractionated bromelain. We have reorganized and abridged the introduction and discussion sections in the study and tried to make it easier to read and catch. 

Q2. The authors state that F3 contains a lectin: did they check the hemagglutination activity of the fraction?

Response: Yes, we published data dealing with Ananas comosus mannose-binding lectin crystal structure and hemagglutination activity (Azarkan et al., Scientific Reports (2018) 8:11508 DOI:10.1038/s41598-018-29439-x). Hemagglutinating assays revealed that this lectin showed no hemagglutination activity of human, rabbit, and sheep erythrocytes. In contrast, the lectin readily hemagglutinates rat erythrocytes.

Q3. MINOR POINTS:

Page 4: please, insert more data about chromatographic runs (for example flow rate and CV for each step).

Page 9 line 28: “chromatographically”.

Figure 1: uL (capitol L)

Figure 2 is very difficult to be read. 

Response: We have added the asked information at page 4 and revised the errors at pages 9 (line 28) and in Figure 1. The quality of Figure 2 was improved for more clarity.

---

## [Decision Letter · Decision Letter 1]

7 May 2023

Cytotoxic Properties of Unfractionated and Fractionated Bromelain Alone or in Combination with Chemotherapeutic Agents in Colorectal Cancer Cells

PONE-D-23-03463R1

Dear Dr. Chang,

We’re pleased to inform you that your manuscript has been judged scientifically suitable for publication and will be formally accepted for publication once it meets all outstanding technical requirements.

Kind regards,

Sreeparna Banerjee, PhD

Academic Editor

PLOS ONE

Additional Editor Comments (optional):

The authors have satisfactorily addressed the comments of both reviewers.

Reviewers' comments:

Reviewer's Responses to Questions

**Comments to the Author**

1. If the authors have adequately addressed your comments raised in a previous round of review and you feel that this manuscript is now acceptable for publication, you may indicate that here to bypass the “Comments to the Author” section, enter your conflict of interest statement in the “Confidential to Editor” section, and submit your "Accept" recommendation.

Reviewer #1: All comments have been addressed

Reviewer #2: (No Response)

2. Is the manuscript technically sound, and do the data support the conclusions?

Reviewer #1: Yes

Reviewer #2: Yes

3. Has the statistical analysis been performed appropriately and rigorously? 

Reviewer #1: Yes

Reviewer #2: Yes

4. Have the authors made all data underlying the findings in their manuscript fully available?

Reviewer #1: Yes

Reviewer #2: Yes

5. Is the manuscript presented in an intelligible fashion and written in standard English?

Reviewer #1: Yes

Reviewer #2: Yes

6. Review Comments to the Author

Reviewer #1: Authors have cleared all queries and manuscript is improved. I would recommend to accept this research paper...

Reviewer #2: The main suggestion during the first Revision was to merge Results and Discussion, in order to make the MS more clear and simple to be read.

However, this modification has not be done.

Moreover, the Authors did not highlighted the various corrections during R1.

7. PLOS authors have the option to publish the peer review history of their article (what does this mean?). If published, this will include your full peer review and any attached files.

Reviewer #1: **Yes: **Nagaraj K

Reviewer #2: No

---

## [Editor Report · Acceptance letter]

23 May 2023

PONE-D-23-03463R1 

Cytotoxic Properties of Unfractionated and Fractionated Bromelain Alone or in Combination with Chemotherapeutic Agents in Colorectal Cancer Cells 

Dear Dr. Chang:

I'm pleased to inform you that your manuscript has been deemed suitable for publication in PLOS ONE. Congratulations! Your manuscript is now with our production department. 

Kind regards, 

on behalf of

Dr. Sreeparna Banerjee 

Academic Editor

PLOS ONE